# Molecular epidemiology of Animal African Trypanosomosis in southwest Burkina Faso

**Robert Eustache Hounyèmè**[1,2], **Jacques Kaboré**[1,3], **Geoffrey Gimonneau**[1,4], **Martin Bienvenu Somda**[1,3], **Ernest Salou**[1,5], **Antoine Abel Missihoun**[2], **Zakaria Bengaly**[1], **Vincent Jamonneau**[4], **Alain Boulangé**[1,4¤] *

**1** Unité de recherche sur les maladies à vecteurs et biodiversité, Centre International of Recherche-Développement sur l'Élevage en zone Subhumide (CIRDES), Bobo-Dioulasso, Burkina Faso, **2** Département de Génétique et des Biotechnologies, Faculté des Sciences et Techniques (FAST), Université d'Abomey-Calavi, Cotonou, Bénin, **3** Unité de Formation et de Recherche Sciences et Techniques (UFR/ST), Université Nazi Boni, Bobo-Dioulasso, Burkina-Faso, **4** INTERTRYP, Univ Montpellier, CIRAD, IRD, Montpellier, France, **5** Institut du Développement Rural (IDR), Université Nazi Boni, Bobo-Dioulasso, Burkina-Faso

¤ Current address: CIRAD, UMR INTERTRYP, Institut Pierre Richet, Bouaké, Côte d'Ivoire
* alain.boulange@cirad.fr

**Data Availability Statement:** The data were deposited on the CIRAD Dataverse (France) https://dataverse.cirad.fr/ and are available at doi.org/10.18167/DVN1/OBRN6E.

## Abstract

### Background

Animal African Trypanosomosis (AAT) is a parasitic disease of livestock that has a major socio-economic impact in the affected areas. It is caused by several species of uniflagellate extracellular protists of the genus *Trypanosoma* mainly transmitted by tsetse flies: *T. congolense*, *T. vivax* and *T. brucei brucei*. In Burkina Faso, AAT hampers the proper economic development of the southwestern part of the country, which is yet the best watered area particularly conducive to agriculture and animal production. It was therefore important to investigate the extent of the infection in order to better control the disease. The objective of the present study was to assess the prevalence of trypanosome infections and collect data on the presence of tsetse flies.

### Methods

Buffy coat, *Trypanosoma* species-specific PCR, Indirect ELISA *Trypanosoma sp* and trypanolysis techniques were used on 1898 samples collected. An entomological survey was also carried out.

### Results

The parasitological prevalence of AAT was 1.1%, and all observed parasites were *T. vivax*. In contrast, the molecular prevalence was 23%, of which *T. vivax* was predominant (89%) followed by *T. congolense* (12.3%) and *T. brucei s.l.* (7.3%) with a sizable proportion as mixed infections (9.1%). *T. brucei gambiense*, responsible of sleeping sickness in humans, was not detected. The serological prevalence reached 49.7%. Once again *T. vivax* predominated (77.2%), but followed by *T. brucei* (14.7%) and *T. congolense* (8.1%). Seven samples, from six cattle and one pig, were found positive by trypanolysis. The density per trap of *Glossina tachinoides* and *G. palpalis gambiensis* was 1.2 flies.

**Funding:** This work was funded by Bill and Melinda Gates Foundation (https://www.gatesfoundation.org/), grant numbers OPP1154033 (JK, GG, MBS, VJ, AB) and INV-001785 (REH, GG, VJ, AB). The funders had no role in study design, data collection and analysis, decision to publish or preparation of the manuscript.

**Competing interests:** The authors have declared that no competing interests exist.

## Conclusions/significance

Overall, our study showed a high prevalence of trypanosome infection in the area, pointing out an ongoing inadequacy of control measures.

## Author summary

In Burkina Faso, like in most countries of sub-Saharan Africa, Animal African Trypanosomosis (AAT) is hampering economic development. It was therefore important to investigate the extent of trypanosome infections after years of control. This study examined circulating trypanosomes in domestic animals using parasitological, molecular and serological tools in southwest Burkina Faso. The prevalence levels observed show that the known epidemiological situation in the region has not really changed. The trypanosome species usually found in the area such as *T. congolense*, *T. vivax* and *T. brucei s.l.* have remained the same, with a stronger presence of *T. vivax*. The low occurrence of tsetse flies and the predominance of *T. vivax* attests to the role of the mechanical vectors in maintaining the disease. Although no cases of *T. brucei gambiense* infection was encountered in the animals examined, trypanolysis tests suggest that there are contact cases in this historical focus of sleeping sickness. Efforts are therefore necessary to reduce or even eliminate the trypanosome burden, and the data provided by this study can assist the decision making.

## Introduction

In Africa, agriculture and livestock are the two main pillars of economic development. Livestock farming in particular is expanding rapidly in West Africa and is the subject of several programs to intensify and improve production systems in order to meet ever-increasing food needs [1]. However, this sector is facing demographic expansion, climate change, and also the strong pressure from vector-borne diseases, that limit its development.

Animal African Trypanosomosis (AAT) or "*nagana*" is arguably the most important disease for ruminants on the continent [2,3]. The impact of trypanosomes on livestock productivity negatively affects millions of people in rural communities who depend on animals for their livelihood [4]. In addition, the same tsetse flies also transmit the agents of Human African Trypanosomiasis (HAT), also known as sleeping sickness [5,6], adding a public health component to an already complicated economic issue [3]. Up to 10 million km$^2$ [7] of potential grazing land in sub-Saharan Africa (nearly 50% of the region's total area), inhabited by nearly 300 million people, is rendered unsuitable for livestock production [8]. AAT also directly affect the number of livestock heads owned by farmers, the composition of the herd in terms of breeds and species, and the grazing pattern. The potential benefits of better control of AAT on the continent, in terms of meat and milk productivity alone, are estimated up to US$700 million per year [2].

AAT is mainly caused by *Trypanosoma congolense*, followed by *T. vivax* and, to a lesser extent, *T. brucei*. While the tsetse fly remains the main vector of trypanosomes, *T. vivax* can also be transmitted mechanically by biting flies, maintaining the infection even in areas where tsetse flies tend to decrease due to anthropisation [9]. In addition, global climate change increases the risk of spread of *T. vivax*, already present in vast areas of Latin America, and recently reported in camels in Iran [10].

The southwest of Burkina Faso is infested with tsetse flies [11]. This region is a historical focus of HAT [12,13] with the last indigenous case detected in 2015, and for which the source

of contamination is unknown. One hypothesis is an animal origin [14]. It is also the best-watered area in the country and thus very favorable for both agriculture and livestock production. The control of this disease has become a vital necessity [15].

For AAT, a progressive control pathway to reduce the disease burden has been proposed [16]. To achieve these goals, intensified disease management and diagnosis is essential, concomitantly to strong political involvement. The existence of reservoirs, latent infections, multiple species and the difficulty of assessing treatment outcomes challenge these efforts. All these means and policies for eliminating AAT will only be useful if there is a good epidemiological understanding of the disease. In this context, the main objective of this study was to assess the prevalence of the disease and to determine the species of trypanosomes circulating in southwestern Burkina Faso. Such studies have been conducted in the region in the past [17,18], but due to the elusive and changing nature of the epidemiological situation and environment, periodic assessments are necessary for effective targeted control.

Due to the lack of specific clinical signs of AAT, indirect diagnostic techniques must be used, usually in conjunction, as none are entirely satisfactory in terms of sensitivity, specificity, or ease of use. To get as precise a picture of the situation as possible, we used parasitological, serological, and molecular techniques. This work also offered the opportunity to compare different diagnostic methods in a given setting, a seldom occurrence, as most epidemiological campaigns generally use one, or at best two, techniques. To our knowledge, this is one of the rare studies that actually used the three sets of techniques. In addition, tsetse fly trapping and species determination was conducted in this study to gain further insight regarding potential temporal variation of the presence of the main AAT vector.

## Materials and methods

### Ethics statement

No ethical declaration is required for the collection of samples from domestic animals for diagnostic purposes or during disease control campaigns. The veterinary services of the regional directorates of animal and fishery resources are known and usually involved in the management of health issues alongside livestock farmers. To this end, the objectives of the study were explained to them. A veterinary officer conducted an awareness-raising tour among farmers and village chiefs a few days before the sampling campaigns. During the survey mission, samples were taken with the agreement of the owners who made their livestock available. An antiseptic was applied after each sampling.

### Study area and sampling sites

The study was carried out in the broad vicinity of three regions in the southwestern part of Burkina Faso (Fig 1): Hauts-Bassins, Cascades and Sud-Ouest.

Sampling took place in April and May 2019, that corresponds to the end of the hot dry season, before the onset of the rainy season. In each area, the villages were chosen in relation to rivers, accessibility of farms, and consent of the breeders for animal blood sampling. Three localities were sampled in the Hauts-Bassins region: Kôlôkô, Kangala, and Orodara; three in the Cascades region: Ouéléni, Soubakaniédougou, and Niangoloko; and three in the Sud-Ouest region: Loropéni, Kampti, and Batié.

### Animal sampling

Sampling was carried out on all domestic animals proposed by the breeders (cattle, sheep, goats and pigs). Thus, in some farms the selection of animals to be sampled was not random

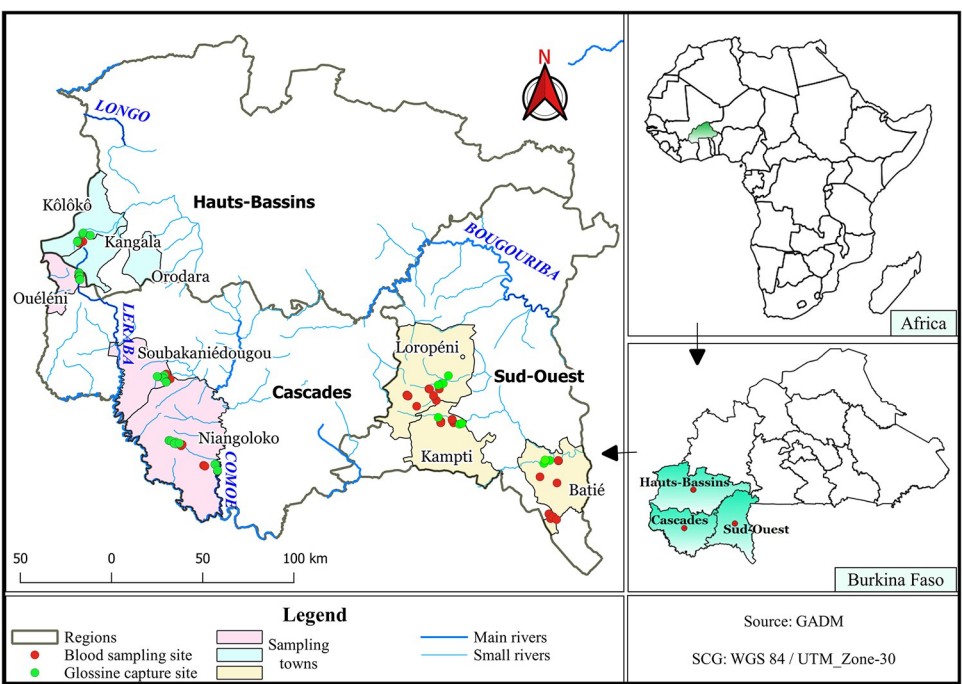

**Fig 1. Map of the study area and geographic distribution of sampling and tsetse capture sites.** The study area is located in the southwest of Burkina Faso, in West Africa. Three localities were sampled in the Hauts-Bassins region: Kôlôkô, Kangala, and Orodara; three in the Cascades region: Ouéléni, Soubakaniédougou, and Niangoloko; and three in the Sud-Ouest region: Loropéni, Kampti, and Batié. Each red circle represents a blood sampling area, either communal corral or household, and each green circle represents a tsetse capture location constituted by three to five traps along riverbanks, 50 m apart. Map generated with QGIS 3.18 (QGIS Development Team, 2018. QGIS Geographic Information System. Open Source Geospatial Foundation Project) based on public geographic data extracted from GADM data 4.0.4 contributors (https://geodata.ucdavis.edu/gadm/gadm4.0/gpkg/gadm40_BFA_gpkg.zip) under CC-BY open license (https://gadm.org/license.html).

but would follow the farmer's choice. In some instance, the selection was skewed towards animals with poor body conditions, or on the contrary towards valuable animals such as drought oxen, introducing an occasional bias. Dogs, usually included in such surveys, are rare in the area and were not sampled in our study.

Biological data for each animal were recorded on a questionnaire-type sheet. The recorded information was: sampling localization, name of farm, species, age classes, sex, and physical condition of the animals noted from 1 to 5 (1- recumbent animal, 2- animal in very poor condition, 3- animal in fair condition, 4- animal in good condition, and 5- animal in very good condition). The packed cell volume (PCV) values and field diagnostic test results were also noted.

A 5 mL blood sample was taken with a heparinized Vacutainer tube from the jugular vein for cattle, goats, and sheep, and *vena cava* for pigs. Tubes were kept on ice all through the various procedures.

## Animal field survey

Two heparinized microcapillaries for each animal were filled with the previously collected blood and centrifuged at 3000 rpm for 5 min. The PCV was measured in duplicate using a microhematocrit reading plate and recorded. Animal with PCV value below 24% were considered to be anemic. For each of the two microcapillaries, the buffy coats (BC) were subsequently collected through breaking off the capillary just under the white blood cells layer and spread

between slide and cover slip, and examined under an optical microscope with a 40 X magnitude objective to search for the presence of trypanosomes [19]. The Vacutainer tubes were centrifuged on site at 13 000 rpm for 5 min. Plasmas were first collected with a 1000 μL micropipette and transferred to individual 1.5 mL microtubes. The BC were collected separately by pipetting the white layer of leucocytes (500 μL), avoiding as far as possible to pipette red blood cells. Samples were stored at -20˚C in a portable freezer until brought back to the lab and moved to a -20˚C freezer.

## Molecular diagnosis

For DNA extraction the 500 μL BC samples were added to a sterile 1.5 mL microtube containing 500 μL of 5% w/v suspension of Chelex-100. The mixture was incubated first in a 56˚C water bath for 1 h and then at 95˚C for 30 min. The tubes were centrifuged at 13,000 rpm for 3 min, the supernatant recovered in new 1.5 mL microtubes and stored at -20˚C.

Determination of trypanosome species was done first using the TRYP1(R & S) primers, which amplify the internal transcribed spacer 1 (ITS1) of the ribosomal DNA of most trypanosome species, and display size polymorphism. Positive samples were subsequently submitted to species-specific primers pairs (Table 1) amplifying satellite DNA of *T. brucei s.l.* (TBR1/TBR2), *T. congolense* "Savannah" type (TCS1/TCS2) [20], *T. congolense* "Forest" type (TCF1/TCF2), *T. vivax* (TVW1/TVW2) [21] and *T.b. gambiense* specific glycoprotein TgsGP [22]. No specific primers were used to detect *T. theileri*, this ubiquitous parasite being considered largely non-pathogenic. All reactions were carried out in a final volume of 25 μL containing 1 X buffer with 2.5 mM $Mg^{2+}$, 0.4 mM of each primer, 0.4 mM of each deoxyribonucleotid, 0.2 units of *Taq* DNA polymerase and 2.5 μL of the undiluted sample to be amplified. The

**Table 1. Primer sequences and amplification product sizes.**

| Type | Specifics | Names and sequences of primers | References |
|---|---|---|---|
| Mono-specific | *T.b s.l.* (177 pb) | TBR 1: 5' CGA ATG AAT ATT AAA CAA TGC GCA G 3'<br>TBR 2: 5' AGA ACC ATT TAT TAG CTT TGT TGC 3' | [20] |
| | *T.v* (150 pb) | TVW_1: 5' CTG AGT GCT CCA TGT GCC AC 3'<br>TVW_2: 5' CCA CCA GAA CAC CAA CCT GA 3' | [21] |
| | *T.cs* (321 pb) | TCS 1: 5' CGA GCG AGA ACG GGC AC 3'<br>TCS 2: 5' GGG ACA AAC AAA TCC CGC 3' | Modified from [20] |
| | *T.cf* (350 pb) | TCF 1: 5' GGA CAC GCC AGA AGG TAC TT 3'<br>TCF 2: 5' GTT CTC GCA CCA AAT CCA AC 3' | [21] |
| | *T.bg* (308 pb) | Tgs-GP F: 5' GCT GCT GTG TTC GGA GAG C 3'<br>TgsGP R: 5' GCC ATC GTG CTT GCC GCT C 3' | [22] |
| Poly-specific | *T.b s.l.* (520 pb) *T.v* (310 pb) *T.cs* and *T.cf* (680–750 pb) *T.l* (623 pb) | TRYP 1S: 5' CGT CCC TGC CAT TTG TAC ACA C 3'<br>TRYP 1R: 5' GGA AGC CAA GTC ATC CAT CG 3' | [23] |

*T.b s.l.*: *Trypanosoma brucei sensu lato*; *T.v*: *Trypanosoma vivax*; *T.cs*: *Trypanosoma congolense* Savannah type; *Trypanosoma congolense* Forest type; *T.bg*: *Trypanosoma brucei gambiense*; *T.l*: *Trypanosoma lewisi*.

amplification program (TRYP1R & S, TBR1/TBR2, TCF1/TCF2, TCS1/TCS2 and TVW1/TVW2) started with an initial denaturation step at 95˚C for 3 min followed by 40 cycles composed of a denaturation step at 95˚C for 30 s, a hybridization step at 55˚C for 1 min 30 s and an extension step at 72˚C for 1 min, and a last final extension at 72˚C for 5 min. The TgsGP PCR was performed using a program consisting of an initial denaturation step at 94˚C for 15 min followed by 45 cycles of a denaturation step at 94˚C for 30 s, a hybridization step at 63˚C for 30 s and an extension step at 72˚C for 30 s. The amplicons were separated by electrophoresis on a 2% agarose gel containing ethidium bromide, and visualized under UV light.

## Serological diagnosis

We used a protocol adapted from those compiled in the Compendium of Standard Diagnostic Protocols for animal trypanosomoses of African origin of the WOAH (founded as OIE) (https://agritrop.cirad.fr/591960/). Briefly, Polysorp plates (Nunc, Roskilde, Denmark) are coated with whole soluble antigen of *T. vivax*, *T. congolense* or *T.b. brucei* diluted at 5 μg/mL in 0.05 M carbonate-bicarbonate buffer pH 9.6 and incubated at 4˚C overnight. The plates are washed with washing buffer (PBS + 0.1% Tween-20) the next day, and each well filled with 150 μL of blocking buffer (PBS-Tween + 5% skimmed milk) and incubated with constant stirring at 37˚C for 30 min. The plasma diluted 1:100 in blocking buffer are deposited in duplicate in the plates and incubated for 60 min at 37˚C. The plates are then washed three times and filled with 100 μL of conjugate diluted in PBS-Tween-20 in each well, and again incubated for 30 min at 37˚C. The plates are washed four times and in each well 100 μL of TMB substrate (2,2',5,5'Tetramethyl-benzidine) is added and incubated for 30 min in the dark. Optical density (OD) measurements were performed at 620 nm with a Multiskan FC Microplate Photometer (Thermo Fisher Scientific, Waltham, MA, United States). The ODs were then expressed as percentage relative positivity (PPR) compared to positive and negative standards. The threshold of positivity for all three species was set to 20%.

The immune trypanolysis test (TL) mainly used for HAT diagnosis was performed with all sampled animal plasma [24,25]. Animal plasma samples were processed with TL using cloned populations of *T.b. gambiense* variant antigen type (VATs) LiTat 1.3, LiTat 1.5 and LiTat 1.6 as previously described [25]. LiTat 1.3 and LiTat 1.5 VATs are supposed to be specific for *T.b. gambiense*, while LiTat 1.6 VAT is expressed in *T.b. gambiense* and *T.b. brucei* [24]. The test is considered positive when the observed lysis reaches a threshold of 50%.

## Entomological survey

Riverine tsetse flies were captured using biconical traps Vavoua at several sites along the riverbanks in the three study areas for 48 hours each. A total of 97 traps were used throughout the survey and positioned every 50 m along the riverbanks in the vicinity of the sampled farms. They were distributed as follows: 25 in Hauts-Bassins, 35 in Cascades, and 37 in Sud-Ouest (Fig 1). The geographical coordinates of each trap were recorded. Tsetse flies were collected daily throughout the survey period. They were then identified by species, counted, and preserved in absolute ethanol. The ADP, apparent densities per trap, was calculated as "the number of tsetse captured per trap per day".

## Data analysis

Comparisons between the prevalence of the different study areas, the animal species, and the sexes, were made using the Chi-square test ($\chi^2$). The comparison of prevalence between the different infections and the association of trypanosome infections with anemia was performed by the *prop* test. The effects of parameters such as sex, age, PCV, animal species, and study

**Table 2. Sampling summary per region, species and sex.**

| Regions | Sex | Number of sampled animals | | | | |
|---|---|---|---|---|---|---|
| | | Cattle | Sheep | Goats | Pigs | Total |
| Hauts-Bassins | M | 118 | 22 | 6 | 4 | 150 |
| | F | 364 | 109 | 16 | 25 | 514 |
| Cascades | M | 143 | 9 | - | 4 | 156 |
| | F | 549 | 56 | - | 13 | 618 |
| Sud-Ouest | M | 182 | 52 | 6 | 41 | 272 |
| | F | 179 | - | - | - | 179 |
| Total | | 1,535 | 248 | 28 | 87 | 1,898 |

areas on the molecular prevalence were analyzed by the generalized linear model (GLM) after testing the normality of the data. All these analyses were performed using R x64 4.0.2. (https://www.R-project.org/). To analyze the levels of concordance of detection of the tests used between parasitological, serological, and molecular data, a Venn diagram was constructed.

## Results

All numerical data used in this study were deposited on CIRAD Dataverse (France) and can be accessed on https://doi.org/10.18167/DVN1/OBRN6E

### Animal survey

A total of 1,898 domestic animals were sampled, comprising 1,535 cattle, 248 sheep, 28 goats and 87 pigs in the three study regions: 664 in Hauts-Bassins, 774 in Cascades and 460 in Sud-Ouest. Table 2 shows the number of individuals from each species in relation with region and sex. Three hundred animals had a PCV of less than 24%, among them 248 (83%) were cattle, 50 (16.7%) sheep and goats and two (0.7%) pigs.

Of the total, 2 animals were recumbent, 1204 were in very poor condition, 577 were in fair condition, 29 were in good condition and 1 was in very good condition. The details per species classified according to their condition are presented in the Table 3.

### Parasitological investigations

The BCT was positive for 20 animals, leading to an overall prevalence of 1.1% (Table 4). The highest prevalence was observed in Hauts-Bassins region (2.56%), with 17 positive cattle, followed by Sud-Ouest (0.43%) and Cascades (0.13%) with two and one cattle, respectively. The difference in parasitological prevalence between the three regions is significant ($\chi^2$ = 22.5, df = 2, P = $1.3.10^{-5}$) with a predominance in Hauts-Bassins. All observed parasites were of *T. vivax* species, and detected only in cattle.

**Table 3. Animal species according to their physical condition.**

| | Recumbent animal | Very poor condition | Fair condition | Good condition | Very good condition | Total |
|---|---|---|---|---|---|---|
| Cattle | 2 | 1158 | 369 | 5 | 0 | 1534 |
| Goats/sheep | 0 | 37 | 181 | 0 | 0 | 218 |
| Pigs | 0 | 9 | 27 | 24 | 1 | 61 |
| Total | 2 | 1204 | 577 | 29 | 1 | 1813* |

*this kind of data were not collected in the field for 85 animals, hence the difference in number with Table 2

**Table 4. BCT, PCR and ELISA results in the three study regions.**

| Regions | Nb of samples | BCT Nb (%) | Confidence Interval | PCR Nb (%) | Confidence Interval | Elisa Nb (%) | Confidence Interval |
|---------|---------------|------------|---------------------|------------|---------------------|--------------|---------------------|
| Hts-Bassins | 664 | 17 (2.56) | [1.95; 3.17] | 80 (12) | [10.8; 13.3] | 339 (51.1) | [47.2; 55.1] |
| Cascades | 774 | 1 (0.13) | $[4.9.10^{-4}; 0.26]$ | 82 (10.6) | [9.5; 11.7] | 375 (48.5) | [44.9; 52.1] |
| Sud-Ouest | 460 | 2 (0.43) | [0.12; 0.74] | 276 (60) | [57.5; 62.1] | 229 (49.8) | [45.1; 54.5] |
| Total | 1,898 | 20 (1.1) | [0.86; 1.34] | 438 (23.1) | [22.4; 24.1] | 943 (49.7) | [47.4; 52] |

Nb: number; BCT: buffy coat technique

## Molecular diagnosis

The overall molecular prevalence assessed by PCR was 23.1% with a total of 438 positive samples. The Sud-Ouest region showed the highest prevalence with 60% followed by Hauts-Bassins with 12% and Cascades with 10.6% with 276, 80 and 82 positive animals respectively (Table 4). The test comparing the prevalence between the three regions is highly significant (X = 466, df = 2, P = $2.2.10^{-16}$), with a strong predominance in Sud-Ouest.

Furthermore, the GLM showed that the Sud-Ouest region positively influenced (Z = 15.8, P< $2.2.10^{-16}$) the molecular prevalence in contrast to the other regions. Similarly, the sheep (Z = -5.71, P = $9.65.10^{-9}$) and pig (Z = -3.41, P = $2.6.10^{-3}$) species tended to decrease it. In contrast, age (Z = -0.68, P = 0.5) and sex (Z = 0.64, P = 0.51) had no influence on it. Finally, the bovine species was the most affected ($\chi^2$ = $1.47.10^3$, df = 3, P = $2.2.10^{-16}$) with 410 infected animals out of 438 (Table 5).

Molecular examination revealed the presence of the three most common trypanosome species. Overall, *T. vivax* alone accounts for 89% of the infections observed, compared to 12.3% for *T. congolense* and 7.3% for *T. brucei*. In details, *T. vivax* was predominant in Sud-Ouest with 98.6% compared to 85% in Hauts-Bassins and 61% in Cascades. *T. brucei* was relatively more abundant in Hauts-Bassins (20%) than in Sud-Ouest (5.8%). No *T. brucei* infection was found in Cascades. PCR also allowed the identification of the two subspecies of *T. congolense*. Indeed, the prevalence of the "Forest" type of *T. congolense* was 37.8% in Cascades, 5% in Hauts-Bassins and 4% in Sud-Ouest. The prevalence of the "Savannah" type of *T. congolense* was 15.9% in Cascades, 2.5% in Sud-Ouest and 1.3% in Hauts-Bassins (Table 6). No *T. brucei gambiense* infection was detected by PCR. This study shows that the livestock in the area is more under pressure from *T. vivax* than from *T. brucei s.l* ($\chi2$ = 49.4, df = 1, P = $2.1.10^{-12}$).

Mixed infections represented 8.9% of the infections observed. *T. vivax* was more often associated with *T. brucei* than with *T. congolense* ($\chi^2$ = 18.2, df = 2, P = 0.00011), representing 5.3% of all infections, that is nearly 60% of all multiple infections encountered. Mixed *T. brucei* /

**Table 5. PCR results regarding trypanosome species according to host.**

| | Nb | *T.b* Nb (%) | *T.cs* Nb (%) | *T.cf* Nb (%) | *T.v* Nb (%) | Nb P (%) |
|---|-----|--------------|---------------|---------------|--------------|----------|
| Cattle | 1535 | 10 (0.65) | 21 (1.37) | 46 (3) | 368 (24) | 410 (26.7) |
| Goats | 28 | - | - | - | 3 (10.7) | 3 (10.7) |
| Sheep | 248 | 6 | - | - | 3 (1.21) | 9 (3.63) |
| Pigs | 87 | 16 | - | - | 16 (18.4) | 16* (18.4) |
| Total | 1898 | 32 (1.69) | 21 (1.11) | 46 (2.42) | 390 (20.5) | 438 (23.1) |

Nb: number of samples; Nb P: number of positive samples by PCR; *T.v*: *Trypanosoma vivax*; *T.b*: *Trypanosoma brucei*; *T.cs*: *Trypanosoma congolense* Savannah type; *T.cf*: *Trypanosoma congolense* Forest type

*Note that among the 16 infected by *T.b* are the same 16 infected by *T.v*

**Table 6. PCR results regarding trypanosome species according to regions.**

| Species | | Regions | | | Total |
|---|---|---|---|---|---|
| | | Hauts-Bassins | Cascades | Sud-Ouest | |
| Total number of infected animals | | 80 | 82 | 276 | 438 |
| (The total is higher than 100% as some animals have mixed infection) | *T. brucei* | 16 (20%) | - | 16 (5.8%) | 32 (7.3%) |
| | *T. congolense* | 4 (5%) | 35 (42.7%) | 15 (5.4%) | 54 (12.3%) |
| | *T. vivax* | 68 (85%) | 50 (61%) | 272 (98.6%) | 390 (89%) |
| | *T.c « forest »* | 4 (5%) | 31 (37.8%) | 11 (4%) | 46 (10.5%) |
| | *T.c « savannah »* | 1 (1.3%) | 13 (15.9%) | 7 (2.5%) | 21 (4.8%) |
| (Proportion of mixed infections) | *T.c.s+T.c.f[*]* | 1 (1.3%) | 9 (11%) | 3 (1.1%) | 13 (3%) |
| | *T.b+T.c* | - | - | - | - |
| | *T.b+T.v* | 7 (8.8%) | - | 16 (5.8%) | 23 (5.3%) |
| | *T.c+T.v* | 1 (1.3%) | 3 (3.7%) | 11 (4%) | 15 (3.4%) |
| | *T.b+T.c+T.v* | 1 (1.3%) | - | - | 1 (0.2%) |
| | Total | 9 (11.3%) | 3 (3.7%) | 27 (9.8%) | 39 (8.9%) |

T.v: Trypanosoma vivax; T.b: Trypanosoma brucei; T.c: Trypanosoma congolense; T.c.s: Trypanosoma congolense "savannah"; T.c.f: Trypanosoma congolense "forest"

[*] *Irrespective of additional infection by other species*

*T. congolense* infections were not detected, while *T. congolense / T. vivax and T. brucei/ T. congolense/ T. vivax* infections appear very seldom with only one case noted for each. Very interestingly, all the pigs detected positive by PCR harbored a mixed infections *T. vivax/T. brucei* (Table 5). This was not observed in other species.

## Serological diagnosis

The overall seroprevalence as determined by indirect-ELISA on whole trypanosome lysate of the three species *T. brucei*, *T. congolense*, and *T. vivax* was 49.7%, *i.e. a* total of 943 positive samples. The Hauts-Bassins region had the highest prevalence at 51.1%, followed by Sud-Ouest and Cascades at 49.8% and 48.5% respectively (Table 4). The test comparing the prevalence of the three regions did not show any significant difference ($\chi^2 = 0.97$, df = 2, P = 0.62).

Indirect-ELISA tests carried out on all plasma samples allowed to detect the three main species of trypanosomes at the set thresholds. Although this test is not species-specific per se, the comparison of the relative OD values obtained on each of the three lysates used as antigen for a given plasma can allow to deduce the species with a 80% accuracy [26]. Overall, of the 943 positive cases, 728 were *T. vivax* infections, while 139 were *T. brucei* infections, 76 were *T. congolense* (Table 7). In details, *T. brucei* was observed in 15.3%, 15.2% and 13.2% of the positive samples in Hauts-Bassins, Cascades and Sud-Ouest respectively ($\chi^2 = 0.6$, df = 2, P = 0.7). *T. congolense* was observed in 2.4%, 9.9% and 13.6% of the positive samples in Hauts-Bassins, Cascades and Sud-Ouest respectively ($\chi^2 = 25.9$, df = 2, P = $2.35.10^{-6}$). Finally, *T. vivax* was

**Table 7. ELISA results regarding trypanosome species according to regions.**

| Species | Regions | | | Total |
|---|---|---|---|---|
| | Hauts-Bassins | Cascades | Sud-Ouest | |
| *T. brucei* | 52 (15.3%) | 57 (15.2%) | 30 (13.2%) | 139 (14.7%) |
| *T. congolense* | 8 (2.4%) | 37 (9.9%) | 31 (13.6%) | 76 (8.1%) |
| *T. vivax* | 279 (82.3%) | 281 (74.9) | 168 (73.4%) | 728 (77.3%) |
| Total | 339 | 375 | 229 | 943 |

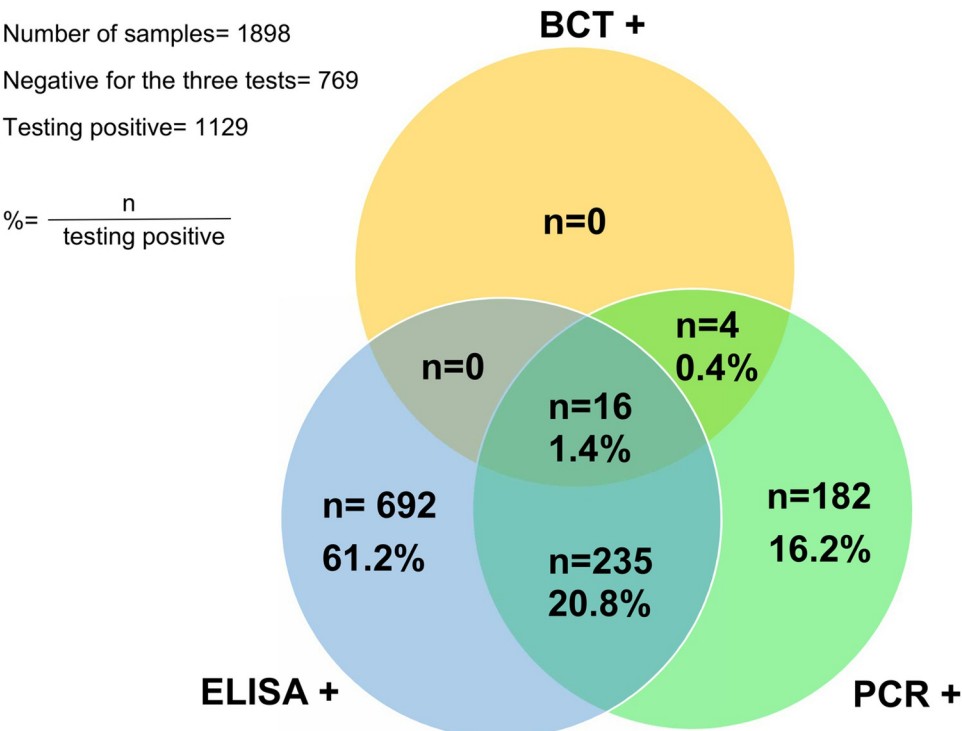

Number of samples= 1898

Negative for the three tests= 769

Testing positive= 1129

$$\% = \frac{n}{\text{testing positive}}$$

**BCT +**

n=0

n=4
0.4%

n=0

n=16
1.4%

n= 692
61.2%

n=235
20.8%

n=182
16.2%

**ELISA +**

**PCR +**

**Fig 2. Concordance diagram for trypanosomes identified by BCT, PCR and ELISA.**

observed in 82.3%, 74.9% and 73.4% of positive samples in Hauts-Bassins, Cascades and Sud-Ouest respectively ($\chi^2$ = 7.7, df = 2, P = 0.02). There is no statistical difference between these prevalence (P> 0.05) for *T. brucei*, very high for *T. congolense*, and borderline for *T. vivax*.

The Venn diagram shows the levels of concordance of detection of the tests used. All BCT positive cases were also PCR positive, however only 16/20 cases were also ELISA positive. Some 57% of cases detected by PCR were also detected by ELISA. Hence, the concordance of cases identified by ELISA and PCR was 22.2% for 248/1129 cases identified (Fig 2).

## Results of trypanolysis

TL performed on 1,828 plasmas revealed seven positive samples, from six cattle and one pig. The pig and one cattle were positive for LiTat 1.3 and 1.5. Four other cattle were positive for LiTat 1.6 and one for all three variants. The Sud-Ouest region had six cases compared to two in Hauts-Bassins.

The unique TL-positive pig was also PCR-positive for *T. brucei s.l.*. None of the TL-positive cattle were PCR-positive for *T. brucei gambiense*. However, five out of the six cattle TL-positive were also positive by indirect-ELISA (stronger on *T. brucei* lysate).

## Association between PCR and PCV results

Of the four domestic animal species tested by PCR in this study, cattle appeared the most affected by trypanosome infections. A total of 248 cattle was recorded as anemic. The prevalence of trypanosome-infected cattle as determined by PCR was 32.3% in anemic cattle and 25.7% in non-anemic.

The average PCV of infected cattle was lower than that of uninfected cattle ($\chi^2$ = 60.5, df = 32, P = $1.7.10^{-3}$) (Table 8). That anemia is positively correlated with trypanosome

**Table 8. PCV versus trypanosome species in cattle.**

| Type of infection | Animal Numbers | Average PCV |
|---|---|---|
| *T. vivax* | 347 | 27.22 |
| *T. congolense* | 39 | 28.31 |
| *T. brucei s.l* | 3 | 24.33 |
| Mixed Infections | 21 | 23.51 |
| Mean positive cases | 410 | 27.28 |
| Mean negative cases | 1125 | 28.54 |
| All cases | 1535 | 28.2 |

infection is corroborated by the GLM analysis of molecular prevalence (Z = -2.06, P = 0.039). The lowest average PCV levels were observed for *T. brucei* infection (but P>0.05). Cattle with mixed infections appear with a lower anemia than those infected with a single species but this observation fails to pass statistical analysis ($\chi^2$ = 26.3, df = 29, P = 0.6) (Table 8).

## Entomological survey

Entomological surveys (Fig 3) carried out in the three regions revealed the presence of two species of tsetse flies, *Glossina tachinoides*, present in Cascades and Sud-Ouest regions but absent in Hauts-Bassins, and *G. palpalis gambiensis*, present in Hauts-Bassins and Cascades regions but absent in Sud-Ouest. The observed abundances remain relatively low with 53 *G. tachinoides* and 180 *G. p. gambiensis* caught over the entire period of the study, for a global trap apparent density (AD) of 1.2. In details, the Hauts-Bassins region presented the highest AD with 2.32 flies caught per trap per day, followed by Cascades and Sud-Ouest regions with 1.23 and 0.42 respectively.

## Discussion

The primary aim of the reported study was to update the prevalence of AAT in the southwestern part of Burkina Faso, which remains highly favourable to livestock farming. The 1.1% parasitological prevalence observed in this study is lower than generally reported in this part of

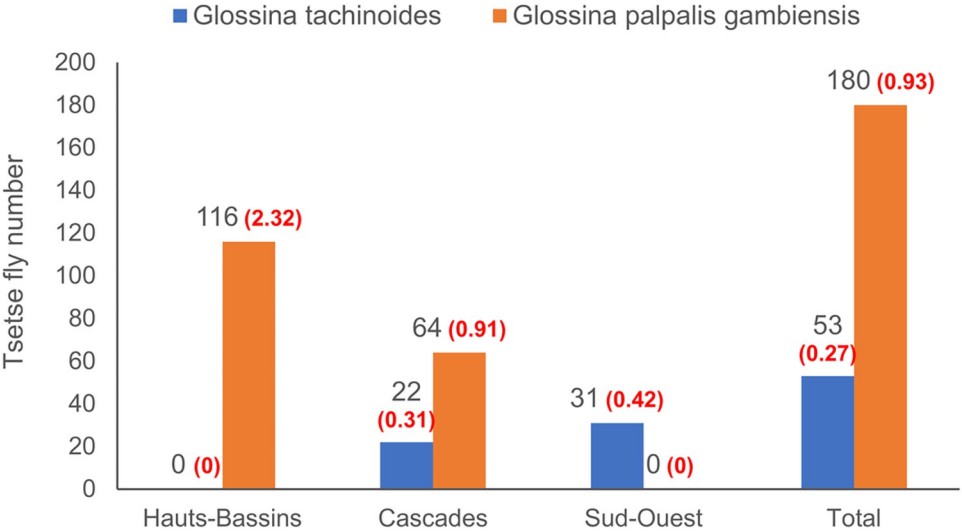

**Fig 3. Results of entomological surveys conducted in the three regions.** In black are the total number of flies caught, in red the global trap apparent density (AD).

the country, where trypanosomosis is endemic. Several studies conducted in the area over the past 20 years have reported parasitological prevalence ranging from 1.67% to 18.3% [17,27]. However, the dry season during which sampling was carried out in our study is known to be very unfavourable for tsetse flies, the main vectors of AAT [28]. The previous studies reporting high prevalence were conducted during either the rainy or the cold season [17]. In addition, they had involved targeted areas in relation to AAT incidence unlike the present study whose localities were representative of the region. It is also possible that infected animals, again in relation to the season in which the study was conducted, are in a chronic phase, more difficult to detect by parasitological techniques [29], especially for *T. vivax* infections. Indeed, these tests often lack sensitivity for the diagnosis of trypanosomosis when the parasitemia is very low (< 300 trypanosomes per mL).

The prevalence as determined by PCR (23.1%) was relatively higher than previous studies in the area. A study in Sidéradougou, in the Northern part of the Cascades region, had showed a molecular prevalence of 11.6%, lower than in the present study [30]. Similar prevalence to ours have been reported in the sub-region by recent studies in Côte d'Ivoire and Niger [31,32], which like Burkina Faso have all experienced intensified tsetse fly control in the past. It would probably be even higher if the treatment with trypanocides was less widespread, as evidenced by the recurrent presence of empty diminazene aceturate packages near the sampling sites. On the other hand, the higher than expected prevalence that we observed may be the result of a bias in the sampling towards animals in poor physical condition.

The seroprevalence of 49.7% that we found in our survey is higher than that of parasitological and molecular prevalence, most likely because circulating antibodies can persist for several months after treatment or spontaneous recovery [33]. Thus, this type of techniques does not diagnose active infections, but rather merely attest to a recent contact of the host with the parasite. This prevalence is still lower than expected in this region. A study reported in 2010 in the Southwest showed a prevalence of 70.7%, significantly higher than that obtained in the present study [17], albeit ten years before, another study carried out in the South Sudanese zone of Burkina Faso showed a prevalence of 43%, closer to the one obtained here [18]. This level of prevalence is also consistent with those reported in some earlier studies in this part of the country [17,18,34] and a recent one in Nigeria [35].

As it happens, *T. vivax* is the predominant species found in this study. This result is in itself particularly interesting, as it offers new insight in the transmission dynamics of trypanosome species. The unusual preponderance of this species can stem from several factors. The fact that this study took place in the dry season, less favorable to tsetse flies, suggests that mechanical transmission of this parasite is, at least in part, responsible for the prevalence observed. Indeed, *T. vivax* can be readily transmitted mechanically by biting flies (*Tabanidae*, *Stomoxys*...) [36], to the point of being widely distributed in Latin America [37]. Still, the survey on bovine trypanosomosis in Sidéradougou in 1997, also carried out in the dry season, showed a preponderance of *T. congolense* [30]. The current low prevalence of *T. congolense* relative to *T. vivax*, besides the capacity of the latter to be mechanically transmitted, may also be due to the level of pathogenicity of this trypanosome, which is generally higher, and the parasitemia less effectively controlled by animals [38]. The severe anemia caused by *T. congolense* can lead to the rapid death of the infected animals. Given that farmers tend to treat only animals that look sick, they are more likely to treat those infected by *T. congolense*, lowering the incidence of this particular species. Still, it is likely that the main factor for the observed *T. congolense* scarcity is the decrease of the number of tsetse flies due to environmental changes, this parasite transmission being strictly cyclical. Altogether, the factors stated above seem to indicate a shift towards *T. vivax* in the recent years. Concerning *T. brucei*, the relatively low prevalence of this species found in our survey is consistent to that reported in other regions of Burkina Faso and beyond, in cattle [39].

The non-detection of *T. brucei gambiense* infection does not exclude the possibility of its presence in the environment. The primers amplify a specific sequence of the *T.b. gambiense* surface glycoprotein (TgsGP) [40]. Although specific for *T.b. gambiense*, it is less sensitive than TBR1/TBR2 PCR because the primers used amplify a unique, unrepeated sequence in the genome. The trypanolysis results show that there is low transmission of *T. brucei s.l.* in the area. This test, designed for HAT, assesses whether a patient has had any contact with *T.b. gambiense*. Although the present study cannot ascertain the presence of *T.b. gambiense*, as the specific PCR test were negative, it is no less true that there are contact cases in the area. In 2015 a last indigenous case was detected in Gouèra (Cascades region). Even if the source of contamination remained unknown, one hypothesis was that it was of animal origin [14]. This may maintain the risk of re-emergence of HAT by animal means as evidenced by the case of the three-variant positive pig [41]. This situation justifies the ongoing surveillance in this area [14].

The hematocrit value is influenced by various factors, mainly nutritional status in an African context, but also the presence of ectoparasites and endoparasites and other infections [42]. However, in tsetse-infested areas, an anemic animal is generally suspected of trypanosomosis [43,44]. In our study, animals infected with one species of trypanosome had lower mean hematocrit than those diagnosed as negative, and animal with multiple infections even lower. Animals harboring *T. brucei s.l* had the lowest mean PCV compared to other trypanosome species. This result is contrary to what is commonly admitted [45], but may still be artefactual given the low numbers of *T. brucei* encountered. In this study as in others [46], there is no significant impact of sex and age on prevalence. However, there could be a difference due to age as farmers often do not take young animals to pasture, which limits their contact with tsetse flies and therefore adults are most exposed [18].

The diagnostic tests that we used in the survey have different levels of sensitivity and specificity. BCT often lacks sensitivity for the diagnosis of trypanosomes when the parasitemia is low ($< 300$ trypanosomes per mL), but is species specific [29]. This explains the low prevalence and single species observed for this test. The sensitivity of the indirect ELISA is high as a pan-trypanosome test, but the species-specificity is greatly reduced as cross-reactions with other trypanosomes occurs. The test is based on a total lysate as antigen, hence the presence of shared antigen determinants between species. Typically, the specificity of an indirect *Trypanosoma sp* ELISA reaches a maximum of 95% [33], unlike PCR which seems to be both more sensitive and specific. The sensitivity of PCR may be reduced for animals in chronic phase [47]. As for trypanolysis, it is still unknown whether the test is specific for *T.b. gambiense* or can cross-react with other *Trypanozoon* [25]. The concordance of the results obtained with BCT and PCR confirms the specificity of these tests for the diagnosis of AAT. On the other hand, there are about 80% false positives determined by ELISA. The difference in specificity between this serological diagnosis and the others is high and its significance could be questioned for the diagnosis of active AAT infections.

The apparent densities of flies per trap observed in relation to the season are low. Entomological surveys in preparation for large-scale tsetse control in 2007–2008 showed higher ADs in the dry season [48]. A longitudinal entomological study would establish the relationship between the period of abundance of cyclic and/or mechanical vectors and trypanosome infections.

In summary, in this survey we have updated the data on the prevalence of trypanosomes in the animal population of southwest Burkina Faso. The result is a better knowledge of African animal trypanosomosis in this part of the country. The study also provides data that will be included in the national atlas of tsetse and AT in Burkina Faso, thus allowing to enhance and update it [49,50]. The spread of *T. vivax* at the expense of *T. congolense*, the likely low-level

presence of *T.b. gambiense*, and the decline of tsetse fly populations are the highlights o1f this report, and should serve as a warning to policy makers.

## Acknowledgments

This study was carried out in collaboration with the Institut de Recherche pour le Développement (IRD), the Centre de Coopération Internationale en Recherche Agronomique pour le Développement (CIRAD) and the Centre International de Recherche-Développement sur l'Elevage en Zone Subhumide (CIRDES). All CIRDES technicians and drivers are thanked for their strong involvement during field sampling and laboratory analysis. We are grateful to the regional and provincial directorates of animal and fishery resources of Hauts-Bassins, Cascades and Sud-Ouest for their participation in this study as well as to all the field actors who were involved in this study.

## Author Contributions

**Conceptualization:** Jacques Kaboré, Geoffrey Gimonneau, Vincent Jamonneau, Alain Boulangé.

**Data curation:** Robert Eustache Hounyèmè.

**Formal analysis:** Robert Eustache Hounyèmè, Geoffrey Gimonneau, Martin Bienvenu Somda.

**Funding acquisition:** Vincent Jamonneau.

**Investigation:** Robert Eustache Hounyèmè, Jacques Kaboré, Martin Bienvenu Somda, Alain Boulangé.

**Methodology:** Robert Eustache Hounyèmè, Jacques Kaboré, Alain Boulangé.

**Project administration:** Geoffrey Gimonneau, Vincent Jamonneau, Alain Boulangé.

**Resources:** Geoffrey Gimonneau, Ernest Salou, Alain Boulangé.

**Supervision:** Jacques Kaboré, Geoffrey Gimonneau, Vincent Jamonneau, Alain Boulangé.

**Validation:** Alain Boulangé.

**Visualization:** Zakaria Bengaly.

**Writing – original draft:** Robert Eustache Hounyèmè.

**Writing – review & editing:** Jacques Kaboré, Geoffrey Gimonneau, Antoine Abel Missihoun, Vincent Jamonneau, Alain Boulangé.

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
