## [Decision Letter · Decision Letter 0]

16 Mar 2022

Dear Dr Boulangé,

Thank you very much for submitting your manuscript "Molecular epidemiology of Animal African Trypanosomosis in southwest Burkina Faso" for consideration at PLOS Neglected Tropical Diseases. As with all papers reviewed by the journal, your manuscript was reviewed by members of the editorial board and by several independent reviewers. In light of the reviews (below this email), we would like to invite the resubmission of a significantly-revised version that takes into account the reviewers' comments. 

As specified by Reviewer 2, all primary data have to be provided in an appropriate format (e.g. a source data file) for transparence and documentation.

We cannot make any decision about publication until we have seen the revised manuscript and your response to the reviewers' comments. Your revised manuscript is also likely to be sent to reviewers for further evaluation.

Sincerely,

Philippe Büscher, PhD

Associate Editor

Michael Boshart

Deputy Editor

Reviewer's Responses to Questions

**Key Review Criteria Required for Acceptance?**

**Methods**

-Are the objectives of the study clearly articulated with a clear testable hypothesis stated?

-Is the study design appropriate to address the stated objectives?

-Is the population clearly described and appropriate for the hypothesis being tested?

-Is the sample size sufficient to ensure adequate power to address the hypothesis being tested?

-Were correct statistical analysis used to support conclusions?

-Are there concerns about ethical or regulatory requirements being met?

Reviewer #1: The manuscript presents a comprehensive epidemiological study of three regions (Hauts-Bassins, Cascades and Sud-Ouest) located in the southwestern part of Burkina Faso, between April and May 2019, at the end of the dry season. The study included 1,898 domestic animals (cattle, sheep, goats and pigs). Various techniques were used to investigate the prevalence of infection of the trypanosomes responsible for AAT, transmitted by tsetse flies: T. congolense, T. vivax and T. brucei brucei. The presence and species of tsetse flies were also assessed. 

The diagnostic methods used in the survey were adequate: PCR with TRYP1(R & S) primers to amplify the ITS1 of the ribosomal DNA of most trypanosome species, which allows species discrimination based on size polymorphism. Species-specific PCR tests were then used on the positive samples: T. brucei s.l. 171 (TBR1/TBR2), T. congolense “Savannah” type (TCS1/TCS2) (20), T. congolense “Forest” 172 type (TCF1/TCF2), T. vivax (TVW1/TVW2) (21) and T.b. gambiense specific glycoprotein 173 TgsGP ”.

Serological diagnosis: whole soluble antigen of T. vivax, T. congolense or T.b. brucei and immune trypanolysis test. 

The entomological survey used biconical traps for tsetse flies at various sites along the riverbanks in the three study areas for 48 hours each. This was followed by species identification. counting and determination of “the number of tsetse captured per trap per day”.

Data analysis and statistical tests were performed to compare: 1. prevalence of the different study areas, the animal species, and the sexes 2. prevalence between the different infections and the association of trypanosome infections with anemia and 3. to study the effects of parameters such as sex, age, PCV, animal species, and study areas on the molecular prevalence. Finally, a Ven diagram was constructed to analyze the levels of concordance of detection of the tests used between parasitological, serological, and molecular data.

Reviewer #2: The methods described in this study are clear and properly referenced. There is however one question that could be addressed by the authors and that is: why was Trypanosoma theileri not included in the study? While being considered mostly non-pathogenic, the presence f this infection can lead to to false positive results in a number of diagnostic approaches. 

Minor comment: could the TRYP1 primers be added to the primer table?

**Results**

-Does the analysis presented match the analysis plan?

-Are the results clearly and completely presented?

-Are the figures (Tables, Images) of sufficient quality for clarity?

Reviewer #1: Results and Discussion: 

1. Table 2 presents a summary of the sampled animals per region, species and sex and n=1898 animals. Table 3 includes the various animal species that were sampled, classified according to their physical condition and n=1813. 

R. Why are the numbers of total samples and animal species in Tables 2 and 3 different? 

In the following tables, which present summaries of BCT, PCR and ELISA results in the three study regions (Table 4) and the same results except for the classification by trypanosoma species (Table 5), 

n= 1898.

R: Aside from the discrimination by trypanosome species, could the authors present the data about possible trypanosome co-infections based on their molecular diagnosis?

This seems relevant, since the authors state in the Discussion (lines 430-432): “In our study, animals infected with one species of trypanosome had lower mean hematocrit than those diagnosed as negative, and animal with multiple infections even lower.

See also reviewer’s next comment about seropositive animals for at least two species of trypanosomes. 

2. Lines 314-317: Concerning the seroprevalence, the authors indicate:

 “Overall, of the 934 positive cases, 527 were T. vivax infections, while 59 were T. brucei infections, 26 were T. congolense and 323 (34.6%) cases were positive for at least two species of trypanosomes (Table 5). 

R. The values (total and %) on Table 5 (%) do not match those of the above mentioned text. 

833 (43.9%) T. vivax, 325 (17.1%) T. brucei and 172 (9,1%) T. congolense.

In addition, there is no data in the table for co-infection with two trypanosome species.

Reviewer #2: While the paper is very clearly written, the result section is very cryptic and most true scientific results are not shown. Hence, the reader is forced to 'accept' the conclusion of the authors, without being given a chance to appreciate the scientific data. Tis is OK for a review, but not really acceptable for scientific 'data' publication.

Question 1: ELISA results in trypanosome research can be 'tricky' to interpret. The OD% explanation in the MM section makes one wonder what the real issue is the readout. This could be anywhere between 0.1 and 4.0 and while a % value will mask any inconveniences, it also makes the data very susceptible to individual interpretations. This reviewer understands that 10 plates of positive result is 'a lot' but the true results should be added as supplementary data. In particular for the 232 double PCR+ results. The reader should be given the chance to see 'what that means'.

The same holds true for the PCR data itself. While this reviewer understands that this is a lot of visual data, it is hard to accept that nothing is shown. For example: a reader could be interested in seeing how a PCR+/ELISA+ sample is scored n gel, versus a PCR+/ELISA- result. This data is important is the work is to be taken as a reference for others, and future studies.

A final questions: the results described in Lines 335- (the TL positive PCR positive T. brucei sample in pig and the T. b. gambiense positive sample in cattle: why are such important results mentioned without showing any true result. The authors must have the visual data in hand, so there is no reason not to show them (on not showing them, without mentioning 'data not shown' is truly not acceptable).

**Conclusions**

-Are the conclusions supported by the data presented?

-Are the limitations of analysis clearly described?

-Do the authors discuss how these data can be helpful to advance our understanding of the topic under study?

-Is public health relevance addressed?

Reviewer #1: Yes, the conclusions are supported by the data presented. The authors clearly state the limitations of the various tests performed, as well as the impact of other variables that may affect the trypanosome prevalence, such as the dry season, during which the sampling was carried out, the bias in the sampling towards animals in poor physical condition.

Yes, the authors addressed the importance of their study to better understand the dynamics of the epidemiology of AAT in Burkina Faso.

Reviewer #2: There are no major issues with the conclusions, other than that they cannot be challenged by the reader (or reviewer) because none of the actual data is shown. In fact, technically most of the 'result' section should be moved to the 'conclusion' section...leaving the actual scientific result/data section virtually empty.

**Editorial and Data Presentation Modifications?**

Reviewer #1: Abstract: 

Lines 31-32. Misspelling: It was therefore important to investigate the extend extent of the infection..

Author abstract: 

Line 59: The low occurrence of tsetse fly 

R: Tsetse flies? 

Introduction: 

Lines 81-82: The impact of trypanosomes on livestock productivity negatively affects millions of people in rural communities who depend on animal for their livelihood

R. Do the authors mean “animal products”? 

Lines 83-84: “adding a public health component to an already pregnant economic issue”

R. Suggestion:” adding a public health component to an already complicated economic issue”

Lines 88-90: The potential benefits of better control of AAT on the continent, in terms of meat and milk productivity alone, are estimated to US$700 million per year (2).

R: are estimated up to US$700 million per year

Lines 448-449 and 450-452: 

The concordance of the results obtained with BCT and PCR confirms the specificity of these tests for the diagnosis of TAA. 

The difference in specificity between this serological diagnosis and the others is high and its significance could be questioned for the diagnosis of active TAA infections.

R: Please, clarify the meaning of TAA, do the authors mean AAT?

Reviewer #2: Data presentation?

See above...virtually no data is shown. Mostly conclusions are presented.

**Summary and General Comments**

Reviewer #1: The study provides a very good assessment of the epidemiology of AAT in Burkina Faso. The authors show excellent command of the strengths, weaknesses and limitations of the various diagnostic tools that were used. The combination and comparison of the various results as well as the statistical analyses performed to compare the data from three different regions, and various species of domestic animals are especially noteworthy and merit publication.

The manuscript requires minor revisions, as indicated above.

Reviewer #2: This is a paper with good potential. All this reviewer is asking for is to show the visual (PCR) and numerical (ELISA) data that has been used to draw the presented conclusions.

PLOS authors have the option to publish the peer review history of their article (what does this mean?). If published, this will include your full peer review and any attached files.

Reviewer #1: No

Reviewer #2: No
---

## [Decision Letter · Decision Letter 1]

22 Jul 2022

Dear Dr Boulangé,

Thank you very much for submitting your manuscript "Molecular epidemiology of Animal African Trypanosomosis in southwest Burkina Faso" for consideration at PLOS Neglected Tropical Diseases. As with all papers reviewed by the journal, your manuscript was reviewed by members of the editorial board and by several independent reviewers. The reviewers appreciated the attention to an important topic. Based on the reviews, we are likely to accept this manuscript for publication, providing that you modify the manuscript according to the review recommendations. 

The original manuscript was revised according to the instructions of the reviewers and is almost ready for acceptance.

Only some minor corrections are suggested by reviewer 1. We trust that the authors will follow these suggestions and will submit a further revised manuscript soon.

Sincerely,

Philippe Büscher, PhD

Academic Editor

Michael Boshart

Section Editor

The original manuscript was revised according to the instructions of the reviewers and is almost ready for acceptance.

Only some minor corrections are suggested by reviewer 1. We trust that the authors will follow these suggestions and will submit a further revised manuscript soon.

Reviewer's Responses to Questions

**Key Review Criteria Required for Acceptance?**

**Methods**

-Are the objectives of the study clearly articulated with a clear testable hypothesis stated?

-Is the study design appropriate to address the stated objectives?

-Is the population clearly described and appropriate for the hypothesis being tested?

-Is the sample size sufficient to ensure adequate power to address the hypothesis being tested?

-Were correct statistical analysis used to support conclusions?

-Are there concerns about ethical or regulatory requirements being met?

Reviewer #1: The manuscript has been extensively revised by the authors and resubmitted. As far as this reviewer is concerned, all the previous comments and observations were addressed. The objectives of the study are clearly stated and justified: 1. to assess the prevalence of various trypanosome infections and 2. to carry out an entomological survey on the presence of tsetse flies in various regions in Burkina Faso. The design of the experiment, ethical aspects, methodology and statistical analysis are appropriate and sustain their conclusions.

Reviewer #2: All previously raised questions have been addressed

**Results**

-Does the analysis presented match the analysis plan?

-Are the results clearly and completely presented?

-Are the figures (Tables, Images) of sufficient quality for clarity?

Reviewer #1: Numerical data have been submitted to a data depository (Dataverse).

Tables 4,5 and 6 were modified in the revised manuscript, which makes it easier to analyze the data. Two tables, 7 and 8 were added. Table 7 presents the serological prevalence of various trypanosomes, measured through indirect ELISA with various soluble trypanosome antigens (T. vivax, T. congolense or T.b. brucei), while Table 8 presents the analysis of PCV values for various trypanosome species. 

However, the following points must be revised/ addressed by the authors: 

Lines 278-280: “The overall molecular prevalence assessed by PCR was 23.1% with a total of 438 positive samples. The Hauts-Bassins region showed the highest prevalence with 60% followed by Sud-Ouest 12% and Cascades 10.6% with 276, 80 and 82 positive animals respectively 280 (Table 4). 

R: The authors must revise and correct either the text or the data in Table 4. If the values presented in Table 4 are correct, the text should read: “The Sud-Ouest region showed the highest prevalence with 60%, followed by Hauts-Bassins with 12% and Cascades with 10.6% with 276, 80 and 82 positive animals respectively (Table 4). 

Line 360. Column heading for Table 8. PCV versus trypanosome species in cattle

R: The column heading “Nbre d’animaux” must be translated to English. 

Lines 368-370: “In details, the Hauts-Bassins region presented the highest AD with 2.32 flies caught per trap per day followed by Sud-Ouest and Cascades regions with 0.42 and 1.23 respectively”.

R: The statement should be corrected to present the regions in decreasing order of AD: 

“In details, the Hauts-Bassins region presented the highest AD with 2.32 flies caught per trap per day, followed by Cascades and Sud-Ouest regions with 1.23 and 0.42, respectively”.

Reviewer #2: All previously raised questions have been addressed

**Conclusions**

-Are the conclusions supported by the data presented?

-Are the limitations of analysis clearly described?

-Do the authors discuss how these data can be helpful to advance our understanding of the topic under study?

-Is public health relevance addressed?

Reviewer #1: The conclusions are supported by the data and the authors clearly discuss the limitations of the study and their epidemiological data. They also highlight the importance of this study and the contribution to implement future control measures and a comprehensive path to control AAT in the region.

Reviewer #2: All previously raised questions have been addressed

**Editorial and Data Presentation Modifications?**

Reviewer #1: Introduction: 

Lines 80-81: “In addition, the same tsetse flies also transmit the agents of Human African Trypanosomosis (HAT), also known as sleeping sickness (5, 6).”

R: Replace the term “Trypanosomosis” for the correct one for human disease: “Trypanosomiasis”. 

“In addition, the same tsetse flies also transmit the agents of Human African Trypanosomiasis (HAT), also known as sleeping sickness (5, 6)”

Reviewer #2: All previously raised questions have been addressed

**Summary and General Comments**

Reviewer #1: The study is novel and comprehensive and draws interesting conclusions as it compares various diagnostic methods and analyzes the prevalence of T. vivax, T. congolense and T. brucei sl, as well as the presence of tsetse flies. The manuscript is well written and greatly improved by the inclusion of numerical data, representative figures of the PCR results and additional tables that facilitate the interpretation of the results.

Reviewer #2: All previously raised questions have been addressed

PLOS authors have the option to publish the peer review history of their article (what does this mean?). If published, this will include your full peer review and any attached files.

Reviewer #1: No

Reviewer #2: No

Figure Files:

Data Requirements:

Reproducibility:

References

---

## [Editor Report · Decision Letter 2]

7 Aug 2022

Dear Dr Boulangé,

We are pleased to inform you that your manuscript 'Molecular epidemiology of Animal African Trypanosomosis in southwest Burkina Faso' has been provisionally accepted for publication in PLOS Neglected Tropical Diseases.

Best regards,

Philippe Büscher, PhD

Academic Editor

Michael Boshart

Section Editor

This second revision of the original manuscript has now been adapted according to all reviewers' comments and is ready for acceptance.

---

## [Editor Report · Acceptance letter]

13 Aug 2022

Dear Dr Boulangé,

We are delighted to inform you that your manuscript, "Molecular epidemiology of Animal African Trypanosomosis in southwest Burkina Faso," has been formally accepted for publication in PLOS Neglected Tropical Diseases.

Best regards,

Shaden Kamhawi

co-Editor-in-Chief

Paul Brindley

co-Editor-in-Chief
